# Green and Efficient Preparation of Tailed Lignin Nanoparticles and UV Shielding Composite Films

**DOI:** 10.3390/nano12152561

**Published:** 2022-07-26

**Authors:** Shiyi Zeng, Shenchong Zhang, Xiaogang Liu, Huifang Zhao, Daliang Guo, Xin Tong, Jing Li

**Affiliations:** School of Environmental and Natural Resources, Zhejiang University of Science and Technology, Hangzhou 310023, China; 212010817001@zust.edu.cn (S.Z.); lhzscex321@126.com (S.Z.); 15033130324@163.com (X.L.); zhf9966@163.com (H.Z.); 08guodaliang@163.com (D.G.); xintong@zust.edu.cn (X.T.)

**Keywords:** tailed lignin nanoparticles, ethanol/water, syringaldehyde, self-assembly, UV shielding

## Abstract

Lignin nanoparticles (LNP) with various morphologies could be prepared with solvent–antisolvent methods. However, the employed toxic chemicals limited its large-scale application. In this study, an extremely green method using only ethanol and water as solvent and antisolvent was reported. Besides, with the syringaldehyde (SA) addition and its anchoring action on the lignin particles, a forming process of the tailed structure was observed and resulted. Moreover, the improved electronegativity originating from the phenolic hydroxyl groups enhanced the size distribution uniformity, and the new absorption peaks at 1190 cm^−1^ demonstrated the involvement of SA in the LNP formation. Lastly, the tailed lignin nanoparticles (T-LNP) composited with, respectively, polyvinyl alcohol, chitosan, cellulose nanofibers, cationic etherified starch, and sodium alginate were successfully prepared. The outstanding UV-shielding and free radical scavenging properties in the above composites showed their great potential in wide applications in packaging materials.

## 1. Introduction

The global goals to achieve a carbon peak and carbon neutrality boosted researchers to exploit the potentialities of green and renewable resources in replacement of fossil-related energies [1,2,3]. As the second most abundant and natural biopolymer, lignin has attracted tremendous interest [3] since the diversity of functional groups provided various properties, such as antioxidant [4], ultraviolet shielding [5], anti-microbial activity [6,7], etc. However, due to the lignin heterogeneity, only 2–5% quantities of lignin have realized commercial applications in the field of composite materials [7], biofuels [8] and so on, while the rest was just burnt as low-value fuel [9]. The present approach not only failed to fulfill the vast high-added value of lignin, but also led to the production of environmentally oxygenated polycyclic aromatic hydrocarbons (PAHs), organic pollutants, and particulate matter [10], which could result in a negative effect on human health and the environment.

The main reason limiting the use of lignin is its complex non-homogeneous structure and inherent poor dispersibility, which is highly influenced by the raw biomass materials and the corresponding delignification processes [11]. Recent studies have shown that the heterogeneity of lignin could be overcome when it was prepared as lignin nanoparticles [1]. So far, various chemical methods as well as their accessibility to prepare lignin nanoparticles have been proved including acid precipitation [12,13,14], solvent–antisolvent [13,15,16,17], solvent exchange/shifting [18,19,20,21,22], and so on. With highly alkaline black liquor as the raw material, under a temperature of 121 °C and a pH of 1–4 by adding sulfuric acid, lignin microspheres with narrow size distribution (0.8–1.0 μm) were prepared and demonstrated its successful application in drug delivery and heavy metal removal [12]. Besides, acetylated kraft lignin in dimethyl sulfoxide (DMSO) [13] and low-sulfonated lignin in ethylene glycol (EG) [14] was employed to form nano-size lignin particle (198–236 nm) through acid precipitation mechanism by diluted HCl aqueous solution, followed by an investigation of their potential applications in drug delivery and antimicrobial agents. Moreover, based on solvent and anti-solvent mechanisms, various types of original lignin, including acetylated kraft lignin [13], kraft lignin [15], enzymatic lignin [16,23], etc., have been employed to form colloidal nanoparticles (45–250 nm), spherical lignin-based colloidal nanoparticles (200–500 nm), spherical nanoparticles (60–200 nm) through EG-water, tetrahydrofuran (THF)-water, DMSO-sodium acetate, and deep eutectic solvents-acetone buffer. The results showed that the survival rate of Escherichia coli protected by lignin nanoparticles was 70% after 15 min of UV irradiation, compared to 5% for those protected by traditional lignin [24]. Further research also proved the mixed solvent of THF/ethanol-water could also help generate colloidal spherical lignin particles with about 200 nm [17]. Besides, the solvent exchange/shifting method based on the self-assembly mechanism with π–π interactions has also been proved as an efficient method to synthesize lignin nanoparticles with diverse morphology and characteristics. For instance, by dissolving enzymatic hydrolysis lignin and alkali lignin in THF [18,19], kraft lignin in dioxane [20], alkali lignin in methanol [21], soda lignin in ethanol [22], soda lignin in deep eutectic solvents [25], respectively, lignin hollow nanospheres (419–566 nm), lignin microspheres (168–439 nm), solid lignin nanoparticles (200–400 nm), nanoparticles with perfect spheres (100–451 nm), colloidal lignin particles (160–380 nm) were obtained after deionized water was introduced into the system via dialysis. As could be seen in above chemical methods, toxic chemical solvents, such as THF, DMSO, dioxane and acetone were used to promise the uniform formation at the cost of extensive industrial application due to the possible residual hazardous solvent.

Lignin nanoparticles possessed excellent UV-shielding and anti-oxidant properties and therefore, could be composited with various components for preparing functional packaging materials, which was in high demand to satisfy the increasingly high standard to keep the inclusion against ultraviolet and oxidation. High-purity lignin (~98%), obtained from elephant grass by two-step extractions with dilute acid and base solution, was employed for preparing lignin nanoparticles through a solvent (acetone) and antisolvent (MillQ water) methods. Lignin nanoparticles showed higher antioxidant activity (RSI of 82) compared to lignin and commercial antioxidants (BHT and BHA) [26]. Uniform colloidal spheres of different sizes were prepared from enzymatically hydrolyzed lignin and organic solvent lignin and then mixed with pure cream to make lignin sunscreen. The results showed that the sunscreen performance of the nanolignin colloidal spheres was better than that of the original lignin [27]. Spherical lignin particles with an average particle size of 736 nm were prepared by a self-assembly method using kraft lignin as the raw material and acetone as the solvent. It was used as a multifunctional additive for polyvinyl alcohol films, and the results showed that the composite films had excellent UV shielding ability [28]. Corncob alkali lignin was used as raw material to form lignin nanoparticles; deep eutectic lignin nanoparticles less than 100 nm were formed by a continuous dissolution and self-assembly process of solvents. These particles were introduced into a biodegradable polyvinyl alcohol matrix to prepare nanocomposite films with good UV-blocking properties [25]. Spherical lignin nanoparticles with an average size of 13 nm were obtained in a T-shaped microchannel reactor and were used as an additive with polyvinyl alcohol (PVA) to form UV shielding composite films. Compared with the original lignin, the addition of LNS could improve the shielding effect of lignin on UV light by 13.3% at 250 nm [29]. As could be seen, the quantities of research have proved the adaptability of lignin nanoparticles in traditional PVA or neutral aqueous base cream, but it still lacks research about its universal suitability for other novel composites.

Fractionation can reduce the non-uniform nature of lignin [30]; in this study, we explored a novel and effective approach to preparing tailed lignin nanoparticles by combining fractionation and self-assembly techniques, in which a novel morphology was formed due to using SA as an additive. All nanoparticles were systematically characterized to elucidate their morphology, size distribution, Zeta potential, polydispersity index and thermal stability, as well as the involved formation mechanism. Besides, to evaluate its UV-shielding and anti-oxidant properties, LNP was composited with various biodegradable components, including polyvinyl alcohol, chitosan, starch, sodium alginate, and cellulose nanofibers with reference to historical reports.

## 2. Materials and Methods

### 2.1. Materials

Original ethanol lignin was obtained from our previous work [31]. Anhydrous ethanol (EtOH), Analytical grade purity polyvinyl alcohol (PVA), chitosan (CH, μ = 1000 cP, degree of deacetylation > 75%) cationic etherified starch (CES, μ = 460–1200 mPa·s, Whiteness ≥ 88%), Sodium alginate (SAL), cellulose nanofibers (CNF, C_-COOH_ = 2.4 mol/g, W = 1.22 wt.%) 2,2-Diphenyl-1-picrylhydrazyl (DPPH) were purchased from Sigma–Aldrich^®^. Analytical grade purity syringaldehyde (SA) was obtained from commercial sources.

### 2.2. Lignin Solubilization and Purification

First, 10 g of original ethanol lignin was dissolved in 200 mL of anhydrous ethanol and stirred continuously at room temperature (300 rpm, 3 h). Subsequently, the soluble and insoluble fractions were separated by vacuum filtration. The former, denoted as OEL (organosolv ethanol lignin), was obtained by rotary evaporation and vacuum drying and used for the preparation of lignin nanoparticles.

### 2.3. Preparation of Tailed Lignin Nanoparticles (T-LNP) by Solvent-Antisolvent Method

SA-EtOH solutions were firstly obtained by dissolving 0, 0.3, 0.6 and 0.9 mg/mL of SA in EtOH at room temperature, then OEL at a concentration of 1.0 mg/mL was dissolved in the above SA-EtOH solutions, sonicated (50 KHz) and stirred (500 rpm) for 10 min, and denoted as 0-OEL, 3-OEL, 6-OEL, 9-OEL. Subsequently, 50 mL of deionized water (2 mL/min) was added dropwise to the mixed solution using a peristaltic pump (BT100-2J, Baoding, China) and the tailed lignin nanoparticles were gradually formed and denoted as 0T-LNP, 3T-LNP, 6T-LNP, 9T-LNP, respectively. Following these steps, stirring for 4 h allowed most of the ethanol in the suspension to evaporate, then the suspensions were introduced into dialysis bags (MWCO: 8000–14,000) and immersed in deionized water (periodically replaced) for 72 h.

### 2.4. Preparation of T-LNP-Based Composite Films

PVA/T-LNP films: A 1.5 wt.% PVA solution was prepared by stirring PVA and water at 90 °C for 3 h. Then, different amounts of T-LNP (0, 1 and 3 wt.%) were mixed with PVA solution with certain ratios (Table 1) and sonicated for 5 min to obtain PVA, PVA/1T-LNP and PVA/3T-LNP composite membranes by solvent casting.

CH/T-LNP films: Chitosan (CH) and water containing 1% *v/v* glacial acetic acid were stirred at 40 °C for 12 h to obtain chitosan solution (0.15% wt./wt.). CH, CH/1T-LNP, and CH/3T-LNP composite films were produced using the same method used to prepare PVA films.

PVA/CH/T-LNP films: For the preparation of ternary composites films, the PVA and CH solution were firstly mixed, and then a specific proportion of T-LNP (1 and 3 wt.% of the PVA/CH mass) was added to the mixed solution as shown in Table 1. The obtained ternary composites were denoted PVA/CH, PVA/CH/1T-LNP, and PVA/CH/3T-LNP, respectively.

CES/T-LNP films: CES and water were stirred at 90 °C for 1 h to obtain a cationic etherified starch solution (1.5% wt./wt.). Then a specific proportion of T-LNP (1 and 3 wt.% of the CES mass, shown in Table 1) was added to the mixed solution. CES, CES/1T-LNP, and CES/3T-LNP composite films were produced by the flow-casting method.

SAL/T-LNP films: A 1.5 wt.% sodium alginate solution was prepared by stirring sodium alginate and water at room temperature for 6 h. Then, different amounts of T-LNP (0, 1 and 3 wt.%, shown in Table 1) were mixed with sodium alginate solution and sonicated for 5 min to obtain SAL, SAL/1T-LNP and SAL/3T-LNP composite membranes. 

CNF/T-LNP films: The CNF was diluted into a 1 wt.% aqueous dispersion system and the diluted suspension was sonicated using a cell crusher for 30 min. Then, different amounts of T-LNP (0, 1 and 3 wt.%, shown in Table 1) were mixed with CNF and sonicated for 5 min. Finally, the sonicated CNF, CNF/1T-LNP, and CNF/3T-LNP suspension was vacuum filtered into a membrane and dried at 40 °C for 18–20 h.

### 2.5. Antiradical Activity of Migrating Substances

The DPPH radical scavenging activity of PVA and chitosan films containing LNP was determined based on historical reports [32]. The films (0.2 g) were cut into small pieces and immersed in 4 mL of methanol for 24 h at room temperature. The supernatant obtained was used to assess the DPPH radical scavenging activity: the methanol extract (1 mL) was mixed with DPPH in methanol (1 mL, 50 mg/L) to obtain a DPPH concentration solution of 25 mg/L. Absorbance values at 517 nm were obtained using a UV-Vis spectrometer (UV-2600, Shimadzu, Japan). A mixed solution of methanol from pure PVA and DPPH methanol was used as a control. DPPH radical scavenging activity was calculated by Equation (1), where *A_sample_* was the absorbance of the sample and *A_control_* was the absorbance of the control.
(1)$$\left(RSA,\%\right)=\left[\frac{A_{control}-A_{sample}}{A_{control}}\right]*100\%$$

### 2.6. Characterization

A scanning electron microscope (SEM, S3400, Hitachi, Tokyo, Japan) was used to analyze the morphology of the samples. The zeta potential (ζ), diameter and polydispersity indexes (PDI) of the samples were measured using dynamic light scattering (DLS, Zetasizer nano-ZS90, Malvern, UK), and the data were collected and analyzed through three-time measurements. Fourier transform infrared spectroscopy (FT-IR, 8400 S, Shimadzu, Kyoto, Japan) analysis was carried out for the functional groups of the samples. The UV-vis spectra of samples were determined on a Shimadzu UV-2600 spectrometer within the range of 200 to 500 nm. Thermogravimetric analysis (TGA) tests were implemented through a thermogravimetric analyzer (TGA, Seiko Extra 6300, Kyoto, Japan). The maximum thermal degradation temperature (T_max_) and weight-loss rate were obtained from the derivative thermogravimetric (DTG), together with the residue at 800 °C.

## 3. Results and Discussion

### 3.1. Lignin Nanoparticles

#### 3.1.1. Microstructure

Typical SEM images of T-LNP obtained from different SA concentrations were shown in Figure 1. Spherical lignin nanoparticles were first obtained in the formation process absent of SA (Figure 1a). However, the tailed structure was gradually observed with increasing SA addition as shown in Figure 1b, which was possibly due to the phased separation between ethanol and water resulting in the formation of emulsified template [33], or the enhanced polarity discrimination of lignin induced by SA. As the SA concentration increased to 0.6 mg/mL, the tailed structure occupied approximately 50% as shown in Figure 1c, in which we also noticed that the tail was possibly formed on the premise of main sphere formation. Based on this assumption, we continued the SA addition to 0.9 mg/mL, and the results shown in Figure 1d proved that there were almost no spherical but all tailed structures.

#### 3.1.2. Proposed Formation Mechanism

Based on the above observation, the proposed mechanism for the preparation of lignin nanoparticles was shown in Figure 2. Firstly, low molecular weight (Low M_w_) OEL with 75.5% yield was obtained by fractionation and purification. Subsequently, the OEL without any further modification was used as feedstock to prepare lignin nanoparticles using ethanol and water as solvent and anti-solvent according to the layer-to-layer self-assembly approach based on π–π interactions (Figure 2i) [34]. With the SA addition, a forming process of the tailed structure was observed in Figure 2ii. In the above two processes, hydrophilic groups in lignin could be ionized after adding water. With the hydrophobic groups, lignin molecules formed a directional arrangement at the interface between water and ethanol, resulting in the ethanol being wrapped (Figure 2iii-b). With a further increase in water content, more and more water molecules penetrate the membrane, resulting in larger quantities of lignin molecules aggregating on the internal surface of the membrane by layer-by-layer self-assembly. Meanwhile, with the anchoring action of syringaldehyde on the lignin particles, the tailed structure formed gradually (Figure 2iii-c) [35]. The complete removal of small nanoparticles and ethanol during dialysis reduces the overall swelling, resulting in the ultimate formation of lignin nanoparticles (Figure 2i-d) and tailed lignin nanoparticles (Figure 2ii-d). It could be also inferred from these results that the morphology of the nanoparticles could be adjusted by different levels of SA addition.

To demonstrate the driving force of lignin self-assembly, we obtained UV spectra of the original OEL samples and T-LNP as shown in Figure 3. The results showed that all the OEL samples had an adsorption peak at 201 nm, and the T-LNP samples at 208 nm, indicating that a red shift occurs for the T-LNP samples in the water solution, which provided direct evidence to support that π–π interactions among the aromatic structures play a crucial role during the preparation process [36,37].

#### 3.1.3. Morphology Analysis

According to the proposed mechanism, the effect of SA addition on the particle diameter, zeta potential, and polydispersity index for the various T-LNP were firstly tracked and the results were exhibited in Figure 4. As shown, the average diameters of all 3T-LNP, 6T-LNP and 9T-LNP exhibited an increase compared to the nanoparticles without the addition of SA, i.e., 0T-LNP. With increasing SA concentration, there was an increase in the average diameter of T-LNP (especially from 6T-LNP to 9T-LNP) possibly due to the growing tail part which led to a detection error since it could be clearly observed that the diameter of the main sphere part shown in Figure 2ii-c, d had a decline while the tail part had a rise compared with Figure 2ii-a. Besides, a comparison of lignin nanoparticles prepared by various representative chemicals was exhibited in Table 2. As shown, not only a completely green method was provided by this work, but also novel morphology with competitive size was prepared. While increasing the concentration of SA from 0.3 mg/mL to 0.9 mg/mL, despite the particle diameter increasing from 279 nm to 293 nm, the PDI decreased from 0.221 to 0.151, which indicated that the suspension system was becoming more and more stable. Besides, we measured the ζ-potential in the system, and the results showed that the ζ-potential increases with the increase in SA, which provided more negative charges for T-LNP and could help prevent the aggregation of lignin nanoparticles, and therefore, maintain good stability in water [38].

#### 3.1.4. Chemical Characteristics

In the spectrum as shown in Figure 5, the hydroxyl stretching vibration of SA was observed at 3288 cm^−1^, which is consistent with the results in the literature [41]. The CH_3_ symmetric stretching vibration of SA was observed at 3033, 2971, 2942, 2867 and 2840 cm^−1^ in the spectrum [42]. Two O–CH_3_ groups vibrations of SA were observed at 2942 cm^−1^ for asymmetric and 2971 cm^−1^ for symmetric stretching, respectively [43]. In the spectra, the C = O stretching vibration of SA was observed at 1673 and 1609 cm^−1^. CH_3_ asymmetric bending vibrations were observed at 1515, 1454, and 1424 cm^−1^, while CH_3_ symmetric bending vibrations were observed at 1369, 1334, and 1278 cm^−1^ [44,45]. O–H in-plane-bending vibrations were observed at 1255, and 1209 cm^−1^, while the O–H out-of-plane bending vibrations were observed at 730, 670, 635, 586, and 528 cm^−1^ respectively [45]. Compared with the spectrum of 0-OEL and T-LNP, new absorption peaks appeared at 1190 cm^−1^ in the spectra of 3T-LNP, 6T-LNP, and 9T-LNP, indicating the involvement of SA in the preparation process.

#### 3.1.5. Thermal Properties

The thermal properties of 0-OEL, SA, and T-LNP in a nitrogen atmosphere were also evaluated as shown in Figure 6a,b and Table 3. As shown, both 0-OEL and 0T-LNP exhibited a wide degradation temperature range from 200 °C to 600 °C, but the latter sample exhibited a comparatively higher degradation rate and residual weight of 42.9% at 800 °C possibly due to the more homogeneous structure in nanoparticles [46]. Meanwhile, it could be seen that the residual weight at 800 °C showed an apparent decreasing trend with the increase in SA. Nonetheless, as shown in Table 3, the maximum decomposition temperature (T_max_) and onset decomposition temperature (T_10%_) of T-LNPs exhibited an increasing trend with the increase in SA, since the phenolic hydroxyl groups from SA could trap the free radicals generated during pyrolysis, and therefore, significantly retard their degradation [47].

### 3.2. T-LNP Based Composite Films

#### 3.2.1. UV Shielding Performance

Lignin nanoparticles have attracted great interest because of their abundant functional groups, such as phenolic, ketone and aromatic structures, which have good UV-shielding ability [29,37]. In our study, lignin nanoparticles were used as multifunctional bio-additives to prepare nanocomposite films by solvent casting method [48]. As shown in Figure 7a–d, compared to the pure samples, all prepared composite films exhibited excellent UV shielding properties in both UVA (320–400 nm) and UVB (290–320 nm) regions. As T-LNP content in films increased, higher UV shielding capacity was exhibited. However, it was noteworthy that the binary composite film had better UV resistance than the ternary composite film at the same addition level (Figure 7d). A detailed comparison of the optical properties was conducted and shown in Table 4. As could be seen, the tailed lignin nanoparticles showed superior transmittance results at both 320 nm (UVB region) and 550 nm (Visible region), especially compared to the historical results [48]. In addition, the nanocomposite film maintained its transparency as shown in Figure 7e. This effect was mainly due to the occurrence of π–π stacking interactions between the aromatic moieties, forming sandwich-type (H-orientation) and head-to-tail (J-orientation) aggregates, which decreased the energy gap for the π–π electronic transition, thus enhancing the absorption efficacy of UV photons. In addition, charge transfer complexes between electron-donating phenolic groups and electron acceptor ortho-quinones moieties could further increase the UV photo-absorbing capacity of LNP [49].

Besides the above-mentioned PVA and CH, we also investigated the UV shielding ability of the tailed lignin nanoparticles in CES, CNF, and SAL which were all frequently used for functional packaging materials. Results in Figure 8a–c showed that the prepared T-LNP not only had general adaptability in various substrates but also showed excellent UV shielding ability even with only 1% addition. By comparison among the binary films as shown in Figure 8d and the detailed information in Table 5 although all the three components displayed excellence in UV shielding, the CES-based composite films occupied the best one. However, for visible transmittance at 550 nm, compared with the results in Table 4, it could be concluded that there were negative effects from T-LNP addition in CES, CNF and SAL based composite films.

#### 3.2.2. Antiradical Activity of Migrating Substances

In order to further investigate the LNP application potential on packaging towards oxidation, the inclusion of scavenging systems was employed [32,50]. DPPH is a stable free radical that is widely used to test the free radical scavenging ability of lignin and is, therefore, employed in this study [51]. Results of migrating substances for different nanocomposite films directly immersed into the methanol solution for 24 h were shown in Figure 9 and Table 6. With pure PVA as control, the RSA value was 0 as expected. With the addition of 1% T-LNP, the absorption value at 517 nm of the PVA/1T-LNP film decreased from 0.499 to 0.139 and the RSA value increased to 72.1%. The scavenging effect on DPPH was more significant (86.6%) when the T-LNP addition was increased to 3 wt.% (PVA/3T-LNP film), which could be also demonstrated from the visual difference shown in the left three samples in Figure 9a, since DPPH could reduce the DPPH radical to the yellow compound diphenyl bitter hydrazine in the presence of antioxidants (i.e., T-LNP).

For pure CH films without the addition of T-LNP, the original scavenging activity against DPPH (20.0%) existed due to the ability of residual free amino groups (NH_2_) to react with free radicals to form stable macromolecular radicals, after which NH_2_ groups formed ammonium (NH_3_^+^) groups to absorb hydrogen ions from the solution [52]. In the membrane containing 3 wt.% T-LNP (CH/3T-LNP), the antioxidant activity was increased more than 4.3 times compared to the CH sample. Notably, the antioxidant activity was more pronounced in the ternary system than in the binary system with higher RSA values (22.2%, 78.4% and 89.0% for PVA/CH, PVA/CH/1T-LNP, PVA/CH/3T-LNP, respectively).

#### 3.2.3. Thermal Properties

Finally, to investigate the effect of T-LNP loading on the thermal degradation behaviour of the resulting composites, TG curves were recorded for the binary and ternary systems (Figure 10 and Table 7). As shown, at T_10_%, the synergistic reaction between PVA and CH was found since the T_10_% of PVA/CH was 222.4, which was higher than both only PVA (215.7) and CH (137.7). Besides, the synergistic effect among PVA/CH/T-LNP was discovered, since the T_50_% results of PVA/CH/1T-LNP (328.7) were superior to both PVA/1T-LNP (322.7) and CH/1T-LNP (285.9). A similar phenomenon was also seen in the results of composite films with 3T-LNP. Moreover, with the increase in LNP addition from 1% to 3%, the results of T_10_% and T_50_% had an apparent enhancement, meaning that LNP could further help improve the thermal ability.

Besides, as we could see in Table 8, there were differential peak values of DTG for different composite films. As summarized and referred to historical reports [16,48,53], the small weight loss in the range of 75–125 °C (Peak 1) was induced by the loss of adsorbed, bound water; the largest weight loss at 250–280 °C (Peak 2) was due to the initial thermal decomposition of PVA and CH; the range of 340–370 °C (Peak 3) corresponded to the weight loss of dehydration of the saccharides rings and the depolymerization of the acetylated and deacetylated units of the polymer [54]; lastly, thermal degradation of by-products generated by PVA resulted in a weight loss at 400–450 °C (Peak 4). Most importantly it could be summarized from Table 8 that the addition of T-LNP efficiently delayed the whole degradation process, and therefore, strengthened the thermal stability of the composite materials.

## 4. Conclusions

Tailed lignin nanoparticles were obtained from organosolv lignin using the solvent (ethanol)-antisolvent (water) method with the assistance of syringaldehyde (SA). During the solvent transfer, special spherical structures were gradually formed by π–π interactions. With the increase in SA concentration, the increasing zeta potential helped generate smaller-sized and more uniformly tailed nanoparticles. Besides, the tailed lignin nanoparticles were successfully composited with various biodegradable materials, including polyvinyl alcohol, chitosan, cellulose nanofibers, cationic etherified starch and sodium alginate, to evaluate their UV shielding performance. Results showed that T-LNP efficiently improved the UV shielding properties while remained the visible transmittance. Lastly, anti-oxidant characteristics based on radical scavenging ability and improved thermal stability also demonstrated its potential application in packaging materials.

## Figures and Tables

**Figure 1 nanomaterials-12-02561-f001:**
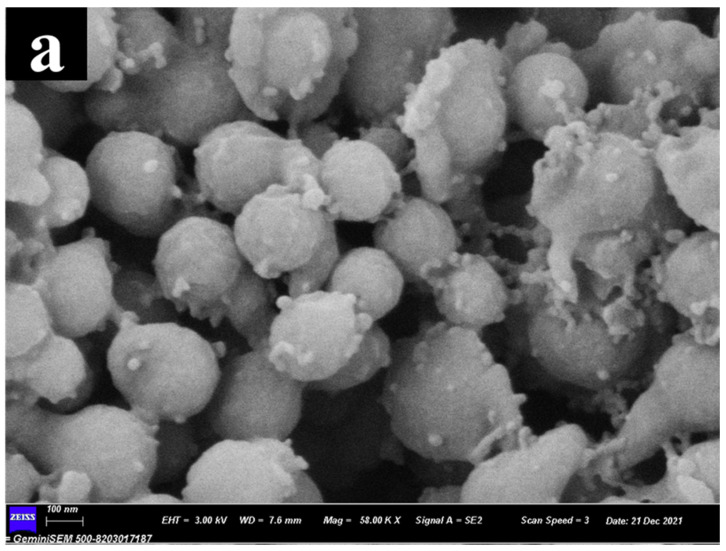
Morphological images of (**a**) 0T-LNP, (**b**) 3T-LNP, (**c**) 6T-LNP, (**d**) 9T-LNP.

**Figure 2 nanomaterials-12-02561-f002:**
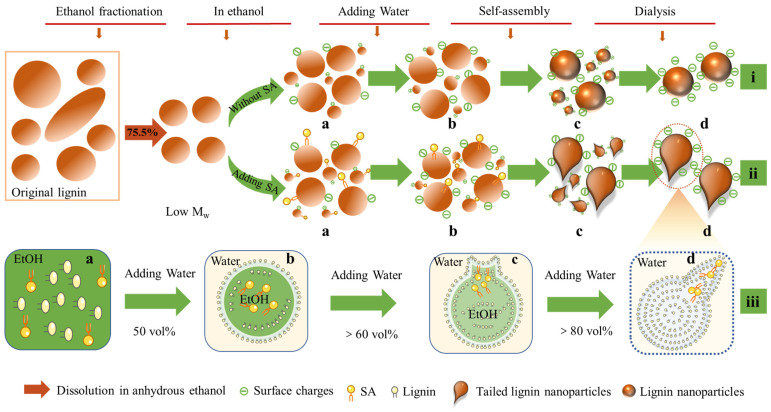
Schematic showing the formation of lignin nanoparticles during the self-assembly process. **Route i:** Using low molecular weight organosolv lignin as raw materials, ethanol and water as solvent and anti-solvent, lignin nanoparticles were obtained by self-assembly method. **Route ii**: Tailed lignin nanoparticles were obtained from low molecular weight organosolv lignin using the solvent (ethanol)-antisolvent (water) method with the assistance of syringaldehyde. **Route iii**: Schematic representation of formation process of the tailed lignin nanoparticles with tunable structures.

**Figure 3 nanomaterials-12-02561-f003:**
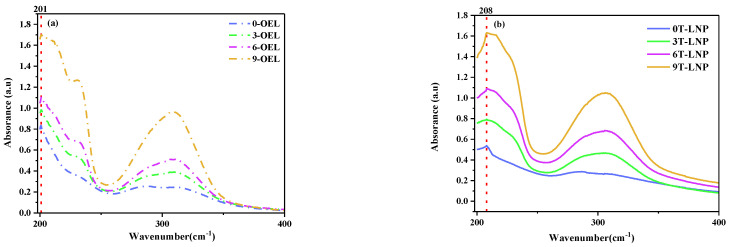
(**a**) UV−vis absorption spectra of OELs and (**b**) T−LNPs.

**Figure 4 nanomaterials-12-02561-f004:**
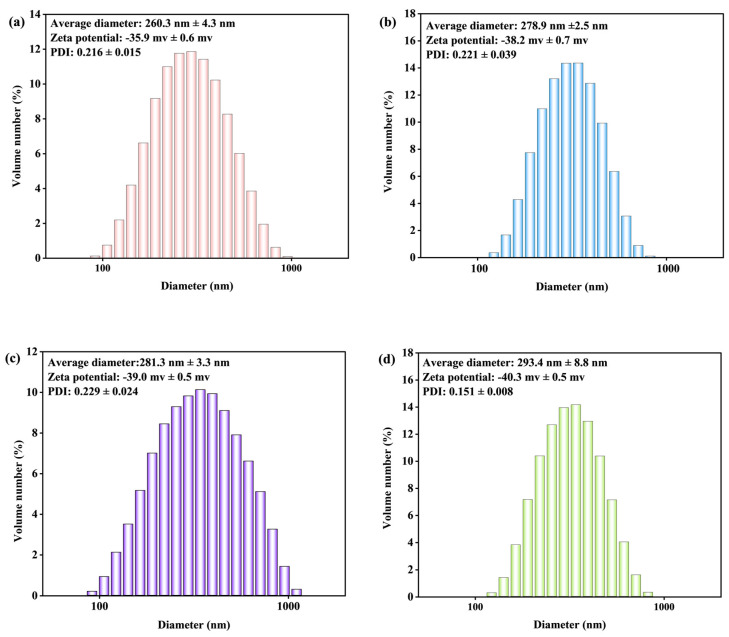
Particle average diameter, ζ, PDI analysis of (**a**) 0T−LNP, (**b**) 3T−LNP, (**c**) 6T−LNP, and (**d**) 9T−LNP.

**Figure 5 nanomaterials-12-02561-f005:**
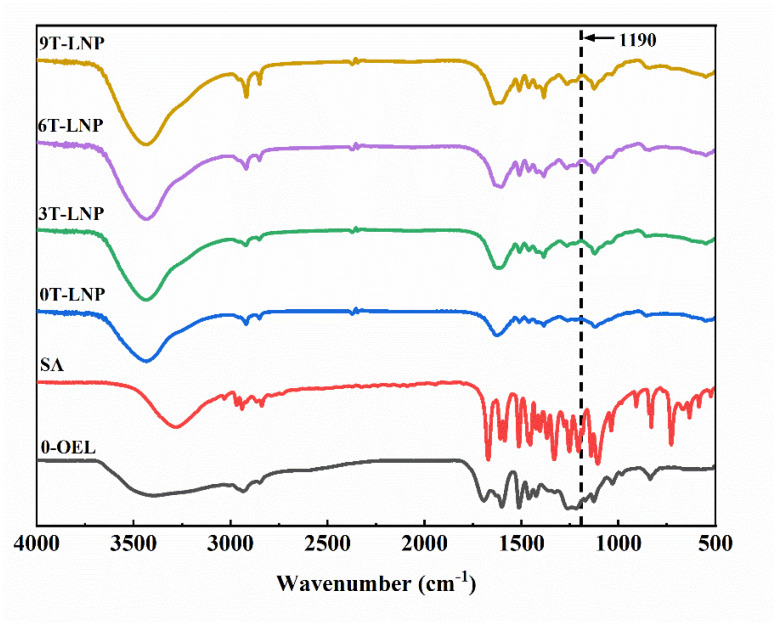
FT−IR spectra of 0−OEL, SA, and T−LNPs.

**Figure 6 nanomaterials-12-02561-f006:**
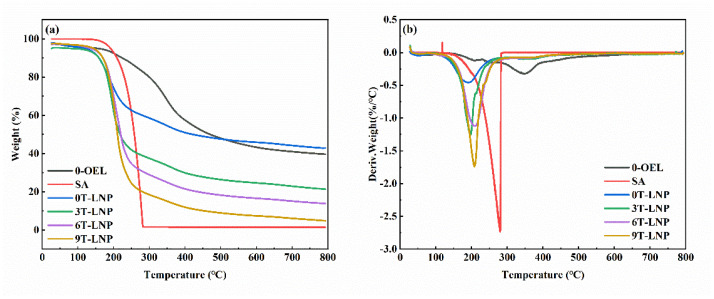
(**a**) TG and (**b**) DTG curves of OEL, SA, and T-LNPs.

**Figure 7 nanomaterials-12-02561-f007:**
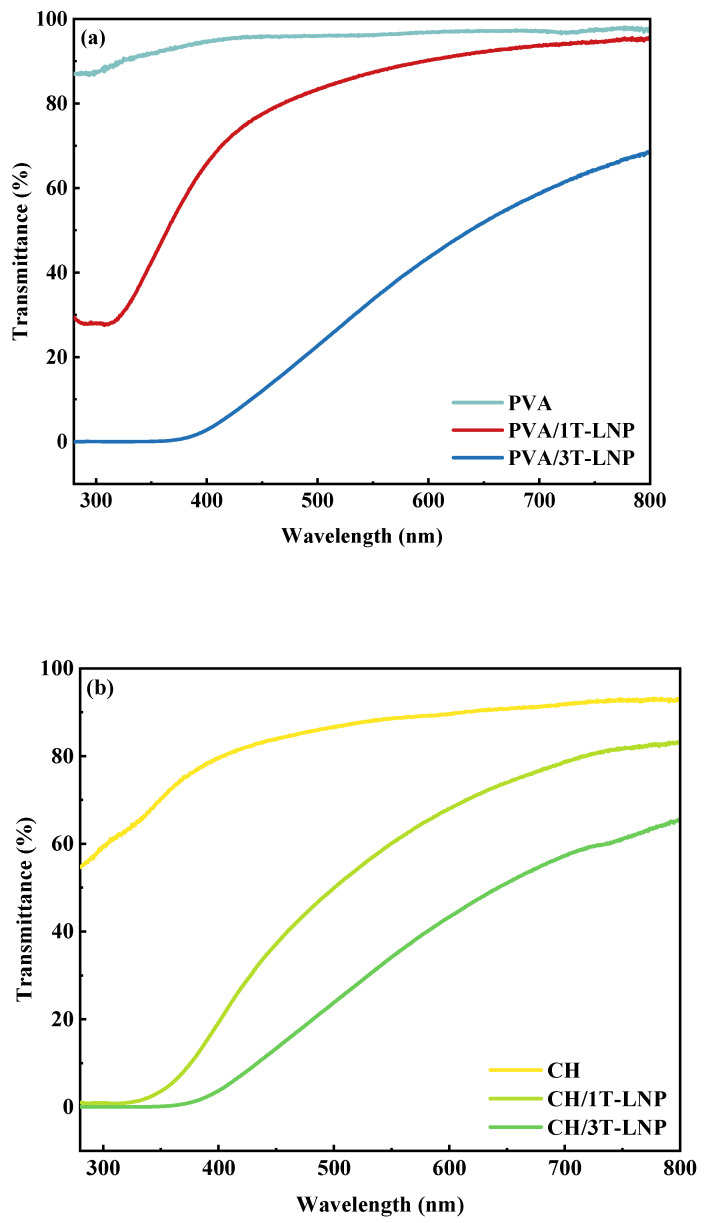
UV–vis spectra of (**a**) PVA, (**b**) CH, (**c**) PVA/CH based composite materials with different amount of LNP; (**d**) UV–vis spectra comparison of the binary and ternary composite films; (**e**) photo demonstration for transparency properties of films obtained from binary and ternary formulations.

**Figure 8 nanomaterials-12-02561-f008:**
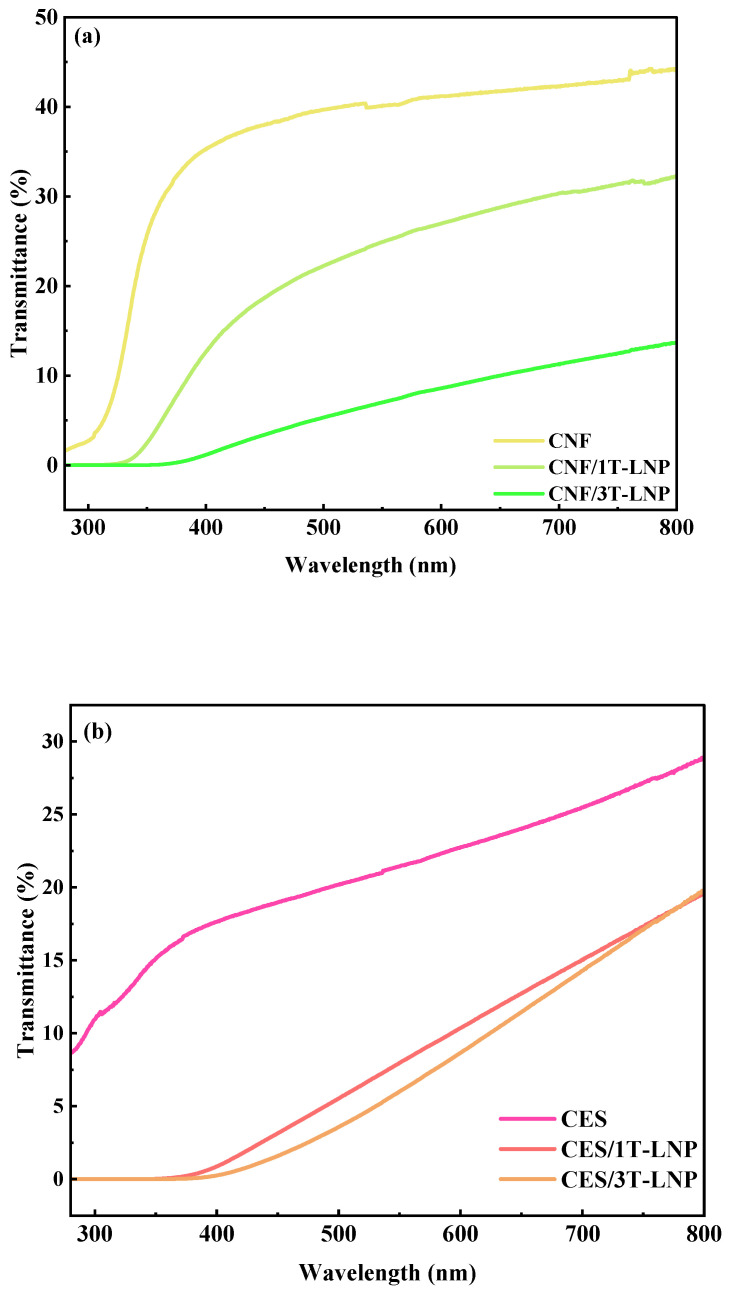
UV–vis spectra of (**a**) CNF, (**b**) CES, (**c**) SAL based composite materials with different amount of LNP; (**d**) UV–vis spectra comparison of the binary composite films.

**Figure 9 nanomaterials-12-02561-f009:**
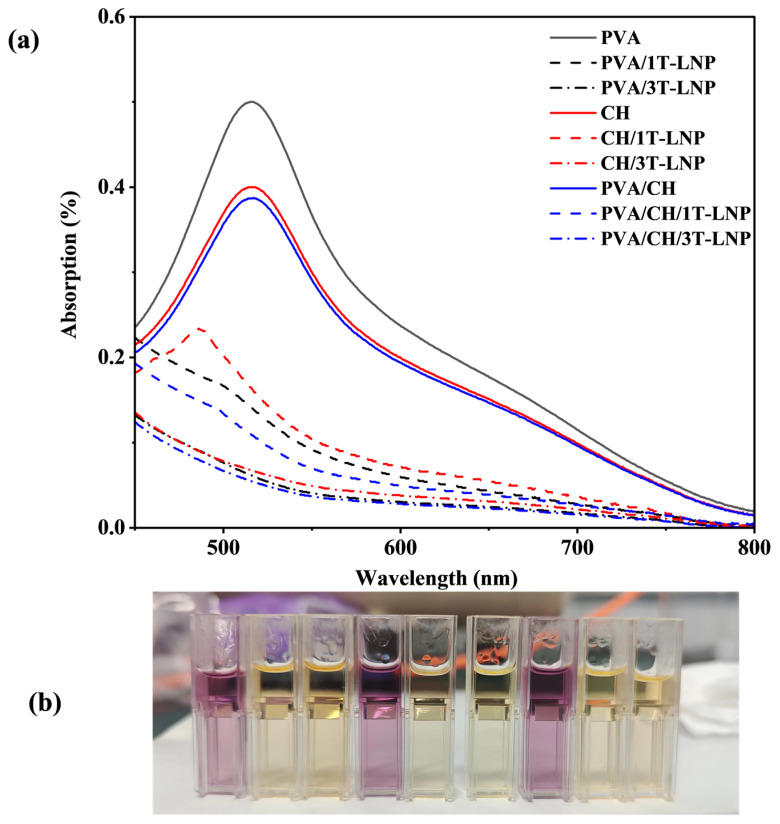
Antioxidant activities of migrating substances for different PVA nanocomposite films immersed directly in the methanol solution for 24 h: (**a**) monitoring of the absorbance for band at 517 nm and (**b**) color variation of the DPPH methanol solution: From left to right: PVA (control), PVA/1T-LNP, PVA/3T-LNP; CH, CH/1T-LNP, CH/3T-LNP; PVA/CH, PVA/CH/1T-LNP, PVA/CH/3T-LNP.

**Figure 10 nanomaterials-12-02561-f010:**
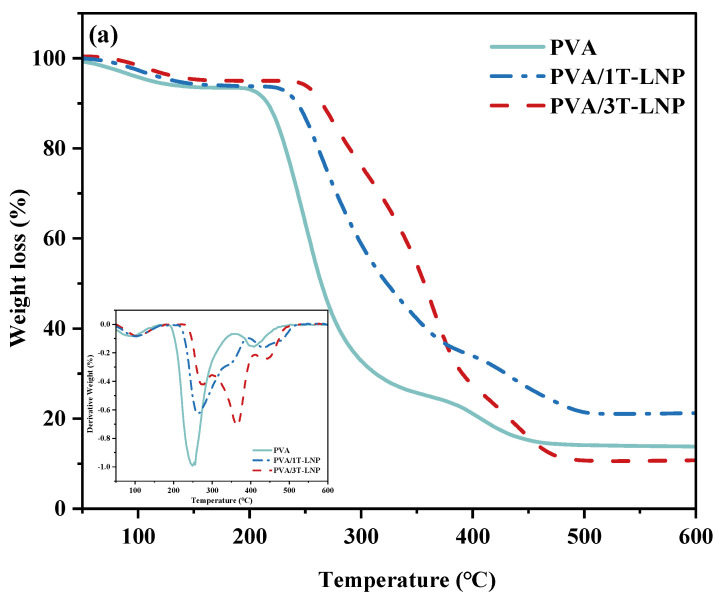
TG and DTG curves of (**a**) PVA, (**b**) CH, (**c**) PVA/CH based composite materials with different amount of LNP; (**d**) Comparison between the binary and ternary composite films.

**Table 1 nanomaterials-12-02561-t001:** LNP-based binary and ternary composite formulations.

	T-LNP (wt.%)	CH (wt.%)	PVA (wt.%)
PVA	-	-	100
PVA/1T-LNP	1	-	99
PVA/3T-LNP	3	-	97
CH	-	100	-
CH/1T-LNP	1	99	-
CH/3T-LNP	3	97	-
PVA/CH	-	10	90
PVA/CH/1T-LNP	1	9.9	89.1
PVA/CH/3T-LNP	3	9.7	87.3
	**T-LNP (wt.%)**	**CNF (wt.%)**	
CNF	-	100	
CNF/1T-LNP	1	99	
CNF/3T-LNP	3	97	
	**T-LNP (wt.%)**	**SAL (wt.%)**	
SAL	-	100	
SAL/1T-LNP	1	99	
SAL/3T-LNP	3	97	
	**T-LNP (wt.%)**	**CES (wt.%)**	
CES	-	100	
CES/1T-LNP	1	99	
CES/3T-LNP	3	97	

**Table 2 nanomaterials-12-02561-t002:** Morphological comparison of lignin nanoparticles prepared by various representative chemicals.

Raw Materials	Methods and Main Chemicals	Morphology	Ref
Enzymatic hydrolysis lignin	Self-assembly, THF-water	Hollow, 419 nm	[18]
Alkali lignin	Self-assembly, THF-water	Sphere, 168 nm	[19]
Kraft lignin	Self-assembly, dioxane-water	Hollow, 200 nm	[20]
Alkali lignin	Self-assembly, methanol-water	Sphere, 100 nm	[21]
Soda lignin	Self-assembly, ethanol-water	Colloidal 160 nm	[22]
Enzymatic hydrolysis lignin	Self-assembly, acetone-water	Sphere, 100 nm	[39]
Organosolv lignin	Self-assembly, THF-water	Sphere, 132 nm	[40]
Ethanol lignin	Self-assembly, ethanol/SA -water	Tailed, 286 nm	This work

**Table 3 nanomaterials-12-02561-t003:** Detailed thermal decomposition parameters for OEL, SA and T-LNPs.

Sample	0-OEL	SA	0T-LNP	3T-LNP	6T-LNP	9T-LNP
TGA T_10%_ (°C)	223.9	209.3	158.7	159.1	169.8	170.1
DTG T_max_ (°C)	345.5	280.1	191.7	197.5	207.9	209.5

**Table 4 nanomaterials-12-02561-t004:** Optical property comparison of prepared PVA, CH composite films with historical reference.

Samples	Optical Property		Ref
	Transmittance (%, 320 nm)	Transmittance (%, 550 nm)	
PVA/3LNP	0.39	38.40	[48]
CH/3LNP	1.52	44.02	
PVA/CH/3LNP	0.87	52.39	
PVA/3T-LNP	0.002	33.64	This work
CH/3T-LNP	0.01	34.11	
PVA/CH/3T-LNP	0.127	58.62	

**Table 5 nanomaterials-12-02561-t005:** Anti-ultraviolet performance comparison of composite films from historical reference.

Samples	Optical Property	
	Transmittance (%, 320 nm)	Transmittance (%, 550 nm)
CES/3T-LNP	0.006	6.04
CNF/3T-LNP	0.001	7.02
SAL/3T-LNP	0.008	6.90

**Table 6 nanomaterials-12-02561-t006:** Antioxidation activity of the migrating substances for PVA, CH and PVA/CH/T-LNP nanocomposites.

Samples	Antioxidation Activity	RSA (%)
	Absorption (λ = 517 nm, %)	
PVA	0.499	0
PVA/1T-LNP	0.139	72.1
PVA/3T-LNP	0.067	86.6
CH	0.399	20.0
CH/1T-LNP	0.159	68.1
CH/3T-LNP	0.066	86.8
PVA/CH	0.388	22.2
PVA/CH/1T-LNP	0.108	78.4
PVA/CH/3T-LNP	0.055	89.0

**Table 7 nanomaterials-12-02561-t007:** Detailed TG parameters for PVA, CH and LNP composited films.

Materials	T_10%_ (°C)	T_50%_ (°C)
PVA	215.7	263.7
PVA/1T-LNP	242.4	322.7
PVA/3T-LNP	264.6	355.2
CH	137.7	366.1
CH/1T-LNP	242.8	285.9
CH/3T-LNP	251.6	303.4
PVA/CH	222.4	316.8
PVA/CH/1T-LNP	225.3	328.7
PVA/CH/3T-LNP	254.9	363.5

**Table 8 nanomaterials-12-02561-t008:** Detailed DTG parameters for PVA, CH and LNP composited films.

Materials	Temperature at Peak (°C)			
	Peak 1	Peak 2	Peak 3	Peak 4
PVA	88.9	255.7	-	406.8
PVA/1T-LNP	97.3	265.1		432.8
PVA/3T-LNP	100.7	272.5	364.2	428.9
CH	77.7	275.7	-	-
CH/1T-LNP	124.8	271.9	-	435.6
CH/3T-LNP	122.1	280.0	-	444.1
PVA/CH	90.7	254.7	347.7	429.3
PVA/CH/1T-LNP	96.9	272.8	353.2	430.7
PVA/CH/3T-LNP	93.8	271.0	360.8	434.9

## Data Availability

The data presented in this study are available on request from the corresponding author.

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
