# Peer review of "Green and Efficient Preparation of Tailed Lignin Nanoparticles and UV Shielding Composite Films"

_nanomaterials, 2022, doi:10.3390/nano12152561_

Round 1
Reviewer 1 Report
I don't see any novelty, so I have to reject it unless you prove novelty.
Reviewer 2 Report
Comments:
Title: “Green and Efficient preparation of tailed lignin nanoparticles and UV shielding composite films”
Journal: Nanomaterials
Manuscript Number: Nanomaterials-1793736
According to Zeng et al, ethanol and water were used as solvents and anti-solvent in the preparation of lignin nanoparticles using the green method. Additionally, syringaldehyde's anchoring effect on lignin particles, as well as the phenolic hydroxyl groups' enhanced electronegativity, improved the uniformity of size distribution. In order to provide UV protection, polyvinyl alcohol and chitosan are combined with the tailed lignin nanoparticles. In spite of this, I believe that the present paper lacks any novelty. As a result, this manuscript should not be published before extensive and significant revisions. It is also necessary for the author to demonstrate some potential applications in the field of packaging or other possible applications.
Several points need to be addressed by the authors.
- There have been several similar works published in the past. It is necessary for the author to review the existing literature.
https://doi.org/10.1080/17458080.2015.1055842
https://pubs.acs.org/doi/10.1021/acsomega.1c02268
https://www.mdpi.com/2076-3921/10/2/274
https://pubs.rsc.org/en/content/articlelanding/2020/GC/D0GC00377H
https://pubs.rsc.org/en/content/articlelanding/2019/ra/c9ra05064g
https://www.sciencedirect.com/science/article/pii/S0141813021009739
- Similarly, many other similar works have been published, so the authors must explain the significance of their work.
- The introduction, results, and conclusion should be revised thoroughly with some new experiments and applications.
- The detailed synthesis and chemistry of the formation of lignin particles should be explained.
- The abstract of the paper mentions the DLS experiment, but no data, analysis, or graph is included. It would be greatly appreciated if you could include the data and analysis.
- It is expected that the authors will provide high-resolution SEM and TEM images. As the SEM image contains only a few particles, it is not possible to determine the size distribution of lignin particles.
- Can you tell me the percentage of lignin particles that are hollow?
- It is necessary to conduct a DSC analysis of different types of lignin particles and to compare the results of the TGA and DTG analyses. Would you be able to add to the analysis and correlate the chemistry behind the stability of lignin particles?
- The addition of a few potential applications is necessary
- There is a lack of scientific insight in the manuscript. It is therefore necessary to provide additional information.
Reviewer 3 Report
In this manuscript the authors propose a green, efficient method for the synthesis of tailed lignin nanoparticles applied in UV shielding materials.
The manuscript provides a detailed characterization of the materials using different techniques (FT-IR, UV-vis spectra TG, DTG,SEM. etc...). This is a very interesting contribution to the use of lignin nanoparticles with novel structure for potential application on UV shielding materials.
The authors should correct the error in the sentence below:
SA-EtOH solutions was were firstly obtained by dissolving 0, 0.3, 0.6 and 0.9 mg/mL of SA 90 in EtOH, and then OEL was dissolved in the above SA-EtOH solution at a concentration 91 of 1.0 mg/mL and denoted as 0-OEL, 3-OEL, 6-OEL, 9-OEL.
The manuscript deserves publication after this minor correction.
Round 2
Reviewer 1 Report
accepted
Author Response
We appreciate your agreement on acceptation.
Reviewer 2 Report
Comments:
Title: “Green and Efficient preparation of tailed lignin nanoparticles 2 and UV shielding composite films”
Journal: Nanomaterials
Manuscript Number: Nanomaterials-1793736
The author of this paper has improved the current version of the paper, however, he did not fully address my questions in the first revision and needs to add some novel insights. In the introduction, please add one more paragraph.
Can you tell me the percentage of lignin particles that are hollow? R7: Good question! Based on the present results, we think that the lignin particles are solid in the sphere part but hollow in the tail part, which will be determined in our further research.
In order to determine whether the particles are hollow or solid, it may be useful to include EDS and TEM analysis of the particles.
I would appreciate if you could provide at least one potential application of this material to offer a novel perspective
Once these three points have been added, the editors may consider accepting this paper for publication if everything appears to be in order.
Author Response
Dear Reviewer:
Appreciate so much for your valuable suggestions, and we have replied to your comments accordingly as attached. If further revisions are needed, please let me know.
We sincerely thank for your review again.
Jing Li
